# Relaxation-Related Piezoelectric and Dielectric Behavior of Bi(Mg,Ti)O_3_–PbTiO_3_ Ceramic

**DOI:** 10.3390/s19092115

**Published:** 2019-05-07

**Authors:** Min Young Park, Jae-Hoon Ji, Jung-Hyuk Koh

**Affiliations:** School of Electrical and Electronics Engineering, Chung-Ang University, Heukseok-Ro 84, Seoul 06974, Korea; pmy730@naver.com (M.Y.P.); hoon2441@naver.com (J.-H.J.)

**Keywords:** piezoelectric, BMT–PT, ceramic

## Abstract

Piezoelectric and dielectric materials have attracted much attention for their functional device applications. Despite its excellent piezoelectric properties, the content of lead in piezoelectric materials should be restricted to prevent future environmental problems. Therefore, reduced lead content in piezoelectric materials with similar piezoelectric properties is favorable. In our research, piezoelectric materials with decreased lead content will be studied and discussed. Even though the lead content is decreased in Bi(Mg_0.5_Ti_0.5_)O_3_–PbTiO_3_ ceramics, they show piezoelectric properties similar to that of lead zirconate titanate (PZT)-based materials. We believe this high piezoelectric behavior is related to the relaxation behavior of Bi(Mg_0.5_Ti_0.5_)O_3_–PbTiO_3_ (BMT–PT) ceramics. In this study, 0.62Bi(Mg_0.5_Ti_0.5_)O_3_–0.38PbTiO_3_ ceramics were prepared by the conventional sintering process. These piezoelectric ceramics were sintered at varying temperatures of 975–1100 °C. Crystallinity and structural properties were analyzed and discussed. X-ray diffraction pattern analysis demonstrated that the optimal sintering temperature was around 1075 °C. A very high Curie temperature of 447 °C was recorded for 0.62BMT–0.38PT ceramics sintered at 1075 °C. For the first time, we found that the origin of the high Curie temperature, d_33_, and the dielectric constant is the relaxation behavior of different dipoles in 0.62BMT–0.38PT ceramics.

## 1. Introduction

Since the discovery of Pb(Zr,Ti)O_3_ (PZT) in the 1950s, it has been applied in piezoelectric transducers, actuators, and sensors because of its excellent piezoelectric properties [1]. PZT is also important in new high-precision switchable measurement methods, where dielectric properties are highly important. These methods compensate for environmental effects, voltage offset, frequency drift, and temperature influence, as previously reported in [2,3,4]. Due to its outstanding piezoelectric and mechanical properties, ceramic-based PZT actuators have received a great deal of attention in the industry. However, it is desirable to use materials with a low lead content to avoid environmental problems [5,6,7,8]. Therefore, there is a great need to discover piezoelectric materials that have a low lead content while still having favorable characteristics like those of PZT ceramics.

Recently, BiFeO_3_-based (BF) perovskite materials have been intensively investigated, owing to their complex electric properties resulting from ferromagnetism and ferroelectricity [9,10,11]. For high temperature device applications, BF-based materials, (1 − x)BiMeO_3_–xPbTiO_3_ (Me^3+^ = Fe, Sc, Mg, In, Y, Yb, Ga), are attractive. These materials contain metallic components in a distorted perovskite structure, which has a higher Curie temperature between the ferroelectric and paraelectric states than that of PZT ceramics [12]. In particular, bismuth-based perovskite system Bi(Mg_0.5_Ti_0.5_)O_3_–PbTiO_3_ (BMT–PT) ceramics are expected to have a higher Curie temperature with relative polarization and piezoelectric charge coefficient. Since the origin of the BiMeO_3_–PTiO_3_ system is multiferroic material, it is expected to have weaker piezoelectric and dielectric properties than the PZT-based system. Despite the higher Curie temperature than that of the PZT system, lower piezoelectric and dielectric properties can be an obstacle for device applications. However, as a lead-containing material, BMT–PT piezoelectric material has similarly excellent piezoelectric properties as PZT, even though it has a low level of lead compared to PZT [13]. The multiferroic and relatively weak ferroelectric properties in the BiMeO_3_–PTiO_3_ system, which is usually observed in the electric field-dependent polarization process, can be the main obstacles for its future piezoelectric applications. However, as we have mentioned before, a relatively high piezoelectric charge coefficient of more than 200 pC/N with a higher Curie temperature range over 400 °C can improve its prospects for future multifunctional applications, including actuators and sensors.

In this study, we will prepare a BMT–PT system by optimizing the sintering temperature. The main advantages of a BMT–PT piezoelectric system are summarized in Table 1. As seen in Table 1, this system has high piezoelectric properties, a high Curie temperature, and a moderate price. However, as a representative lead content-reduced material, BiScO_3_–PbTiO_3_ also has a high piezoelectric coefficient and a high Curie temperature, but this material is very expensive and not feasible for application in electronic devices. The optimized composition of the BMT–PT system was selected through the phase diagram, and then the morphotropic phase boundary (MPB) was extracted to achieve the maximum piezoelectric properties [14]. In our assumption, we believe that 0.62Bi(Mg_0.5_Ti_0.5_)O_3_–0.38PbTiO_3_ ceramics have a mixture of rhombohedra and tetragonal structures. This assumption will be tested and discussed after X-ray diffraction (XRD) analysis. Therefore, the composition of 0.62Bi(Mg_0.5_Ti_0.5_)O_3_–0.38PbTiO_3_ was selected in this experiment [15]. Crystalline properties, including XRD patterns, and electrical properties, including piezoelectric and dielectric properties, will be investigated and discussed.

## 2. Materials and Methods

The 0.62Bi(Mg_0.5_Ti_0.5_)O_3_–0.38PbTiO_3_ (0.62BMT–0.38PT) ceramics were prepared by a standard ceramic sintering technique using the blended oxides method. The raw materials, Bi_2_O_3_, 4MgCO_3_∙Mg(OH)_2_∙5H_2_O, TiO_2_, and PbO, were weighed and combined by ball-milling with a ZrO_2_ ball in ethyl alcohol for 24 h. The blended powders were calcined at 900 °C for 2 h in a furnace. The desiccated powders were blended with polyvinyl alcohol (PVA) and compressed into a disk with a diameter of 10 mm and thickness of 1 mm. The sample was then sintered at various temperatures ranging from 975 to 1100 °C (975, 1000, 1025, 1050, 1075, and 1100 °C) for 2 h in a sealed alumina crucible to avoid loss of Bi_2_O_3_ and PbO due to sublimation. Silver electrodes were formed on both sides of the ceramic plates by screen printing. The poling process was performed in a silicon oil bath at 100 °C with an applied electric field of 1 kV/mm. The crystalline properties were analyzed by XRD using a Cu Kα radiation source (Bruker ARS), and the electric properties were analyzed by an HP 4294 impedance analyzer. Field emission scanning electron microscopy (FESEM) was used to examine the microstructure. To characterize its piezoelectric and dielectric properties, the piezoelectric charge coefficient and electric field-dependent polarization processes were performed, in this experiment, by employing the Sawyer–Tower method with 0.1 Hz.

## 3. Results and Discussion

Figure 1 shows the XRD patterns for the 0.62BMT–0.38PT systems, which were sintered at various temperatures. The specimens were sintered from 975 to 1100 °C in steps of 25 °C. Figure 2 displays the (001) XRD peaks of 0.62BMT–0.38PT piezoelectric ceramics according to the sintering temperature range. As shown in Figure 1 and Figure 2, the XRD patterns show very weak variation in peak intensities and positions. This means that the degree of crystallization and d-space of lattice parameters were slightly changed after the sintering process. As observed in Figure 2, the (001) peak position of 0.62BMT–0.38PT ceramics were shifted to lower angles as the sintering temperature increased. It means that the lattice parameter c increased as the sintering temperature increased. Furthermore, it seems that 0.62BMT–0.38PT piezoelectric systems have good crystalline structure without a pyrochlore phase.

Figure 3 depicts the degree of crystallization for the (001) direction in the 0.62BMT–0.38PT ceramic. The crystallization degree of (001) approached or even exceeded 10% at the sintering temperature of 1075 °C. This means that the distorted perovskite structure of 0.62BMT–0.38PT ceramics has a high degree of c-axis orientation as sintering temperature increases, which corresponds to the added thermal energy to form the crystalline structure. However, when the sintering temperature was increased beyond 1100 °C, the peak ratio of (001) decreased. We believe this maximum value of peak ratio (001) at 1075 °C and decreased peak ratio at 1100 °C are probably related with the crystallization degree. Therefore, we can argue that a sintering temperature of 1075 °C is the optimized sintering temperature owing to the increased (001) relative intensities.

Figure 4 shows the FESEM images for the 0.62BMT–0.38PT ceramics, which were sintered at various temperatures. As shown in the FESEM images, the grain size of 0.62BMT–0.38PT ceramics increased as the sintering temperature increased. At a sintering temperature of 1075 °C, 0.62BMT–0.38PT ceramics showed a large grain size with a dense structure. Due to this highly dense structure, 0.62BMT–0.38PT ceramics that were sintered at 1075 °C showed the highest piezoelectric and dielectric properties among the specimens. However, as the sintering temperature of 1100 °C was attained, the grain shape was distorted, and the size was decreased.

Figure 5 reveals the bulk and theoretical density of 0.62BMT–0.38PT ceramics depending on the sintering temperature. As the sintering temperature increased, the density increased up to 1075 °C and then decreased. This behavior is similar to that of grain size as observed in the FESEM images. As the sintering temperature increased, 0.62BMT–0.38PT ceramics became crystallized, therefore, the density was increased. However, as the sintering temperature was further increased up to 1100 °C, the density decreased. We believe that this decreased bulk density comes from the oversintered phase of 0.62BMT–0.38PT ceramics. It also seems that the bulk density of 0.62BMT–0.38PT ceramics correlated with the FESEM images, which is described in Figure 4.

Figure 6 shows the frequency-dependent dielectric permittivity of 0.62BMT–0.38PT ceramics from 1 kHz to 1 MHz. Clearly, 0.62BMT–0.38PT ceramics sintered at the 1075 °C showed the highest dielectric permittivity of 625 at 1 kHz, whereas those sintered at 975 °C showed the lowest dielectric permittivity of 480 at 1 kHz.

It seems that 0.62BMT–0.38PT ceramics have high dielectric permittivity ranges at room temperature. The solid lines indicate the simulation fitting from the measured samples with the power law. The power law of ε_r_ = *Af*^−n^ was employed to fit dielectric relaxation behavior [21]. Ceramics of 0.62BMT–0.38PT sintered at 1075 °C showed the highest exponent of 0.219 among the specimens. Compared to other specimens, this means that the variance of dielectric permittivity for 0.62BMT–0.38PT ceramics sintered at 1075 °C has the highest exponent of 0.219 and highest dielectric permittivity of 625. The high exponent value of 0.219 from the power law equation means that many different dipoles were involved in the relaxation process with continuous freezing out of dipoles with increasing frequency range. Therefore, 0.62BMT–0.38PT piezoelectric ceramics showed rapid change in the dielectric constant when increasing the frequency range. Owing to this high relaxation behavior, 0.62BMT–0.38PT ceramics sintered at 1075 °C have the highest piezoelectric properties. The dielectric permittivity of 0.62BMT–0.38PT ceramics sintered at 1075 °C also have the highest dielectric permittivity values.

Figure 7 shows the electric field-dependent polarizations for 0.62BMT–0.38PT ceramics with different sintering temperatures. As the sintering temperature was increased, the polarization increased. In particular, 0.62BMT–0.38PT ceramics sintered at 1075 and 1100 °C showed higher ferroelectric properties compared to other specimens. It seems that the higher the sintering temperature, the larger the polarization behavior. In the case of the electric field-dependent polarization case, sintering temperature dependencies are more clearly observed.

Figure 8 shows the piezoelectric charge coefficient and electromechanical coupling coefficient of 0.62BMT–0.38PT ceramics depending on the sintering temperature range. The piezoelectric charge coefficient and electromechanical coupling coefficient of 0.62BMT–0.38PT ceramics sintered at 1075 °C showed the highest values of 240 pC/N and 47%, respectively. These high piezoelectric charge coefficient and electromechanical coupling coefficient coincided well with the dense behavior observed in FESEM images and bulk densities.

Figure 9 displays the piezoelectric voltage coefficient and figure of merit (FoM) of 0.62BMT–0.38PT ceramics. As shown in Figure 9, the highest value of the piezoelectric voltage coefficient of 41 × 10^−3^ Vm/N and FoM of 9.8 pm^2^/N were obtained for the 0.62BMT–0.38PT ceramics. Since the piezoelectric voltage coefficient was derived from the piezoelectric charge coefficient and dielectric constant, g_33_ = d_33_/ε_r_, we can expect that the piezoelectric voltage coefficient of 0.62BMT–0.38PT ceramics sintered at 1075 °C has the highest values of 41 × 10^−3^ Vm/N and FoM of 9.8 pm^2^/N.

Figure 10 shows the temperature-dependent dielectric permittivity ε_r_ of the 0.62BMT–0.38PT ceramic sintered from 975 to 1100 °C. At sintering temperatures of 975, 1000, 1025, 1050, 1075, and 1100 °C, the Curie temperature was 346.9, 374.5, 386.5, 389.2, 447.2, and 428.5 °C, respectively. As the sintering temperature increased from 975 to 1075 °C, the Curie temperature increased from 346.9 to 447.2 °C. As the sintering temperature reached 1100 °C, the Curie temperature suddenly decreased. Compared to other specimens, BMT–PT ceramics sintered at 975 °C had the lowest Curie temperature of 346.9 °C, whereas those sintered at 1075 °C had the highest Curie temperature of 447.2 °C.

## 4. Conclusions

In this research, the piezoelectric and dielectric properties of 0.62BMT–0.38PT ceramics with different sintering temperatures were investigated and discussed. A very high Curie temperature of 447 °C was recorded for 0.62BMT–0.38PT ceramics sintered at 1075 °C. 0.62BMT–0.38PT piezoelectric ceramics have a reduced lead content and high piezoelectric charge coefficient of 237 pC/N and can, therefore, be used in piezoelectric applications where the environment is harsh. Even though 0.62BMT–0.38PT ceramics had a lower lead content compared to Pb(Zr_0.5_Ti_0.5_)O_3_-based conventional piezoelectric ceramics, 0.62BMT–0.38PT piezoelectric ceramics still have excellent piezoelectric and dielectric properties with a high Curie temperature. We believe that the high piezoelectric properties of 0.62BMT–0.38PT are related to relaxation behavior. We believe 0.62BMT–0.38PT ceramics, with their lower lead content, can act as alternative materials for lead-based piezoelectric materials. Reducing the lead content of piezoelectric materials is expected to help not only by preventing environmental pollution on Earth but, also, by being more useful as a piezoelectric material because of its excellent characteristics compared to lead-free piezoelectric material. For example, 0.62BMT–0.38PT ceramics can be applied in piezoelectric transducers or as a sintering aid, using ferroelectricity, that does not change at a high temperature.

## Figures and Tables

**Figure 1 sensors-19-02115-f001:**
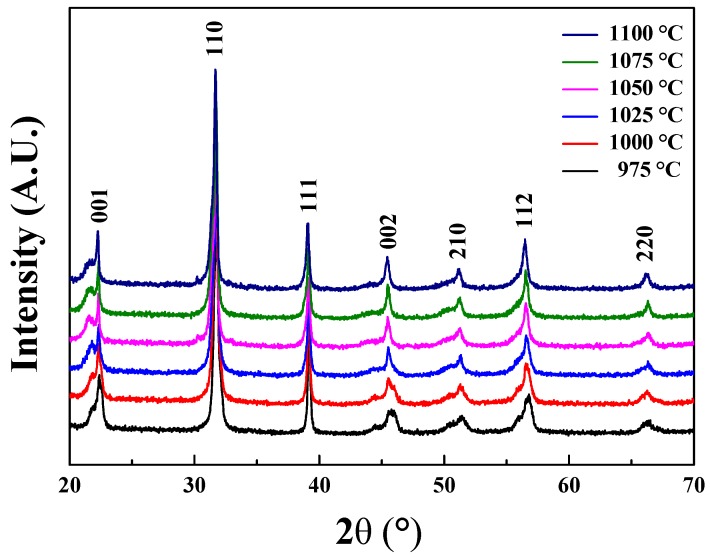
X-ray diffraction (XRD) patterns of Bi(Mg_0.5_Ti_0.5_)O_3_–PbTiO_3_ ceramics according to sintering temperature.

**Figure 2 sensors-19-02115-f002:**
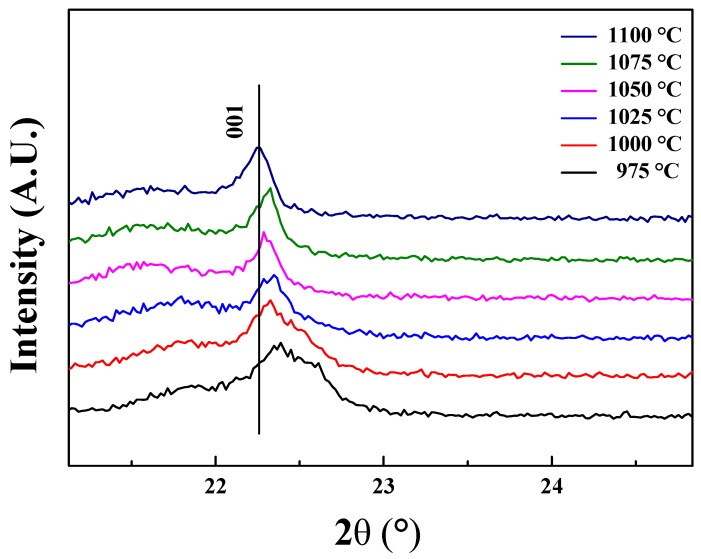
XRD (001) peak shifts of Bi(Mg_0.5_Ti_0.5_)O_3_–PbTiO_3_ ceramics according to sintering temperature.

**Figure 3 sensors-19-02115-f003:**
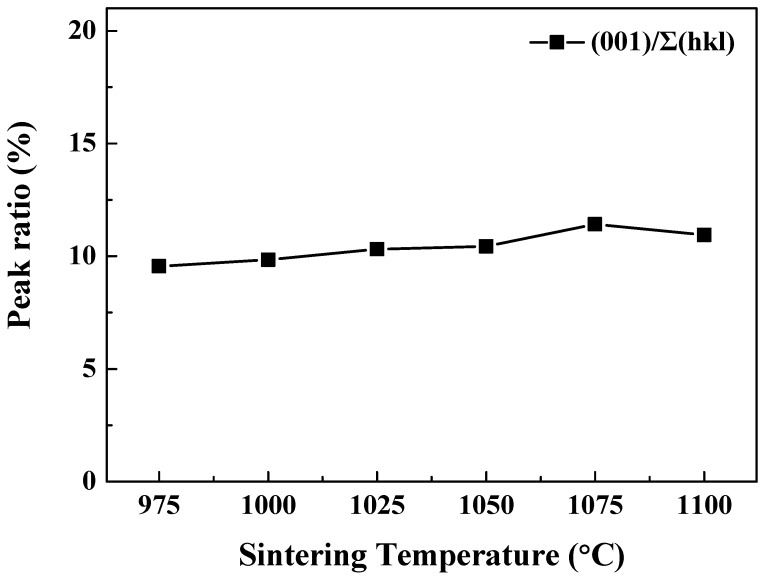
Peak ratio (001) of Bi(Mg_0.5_Ti_0.5_)O_3_–PbTiO_3_ ceramics according to sintering temperature.

**Figure 4 sensors-19-02115-f004:**
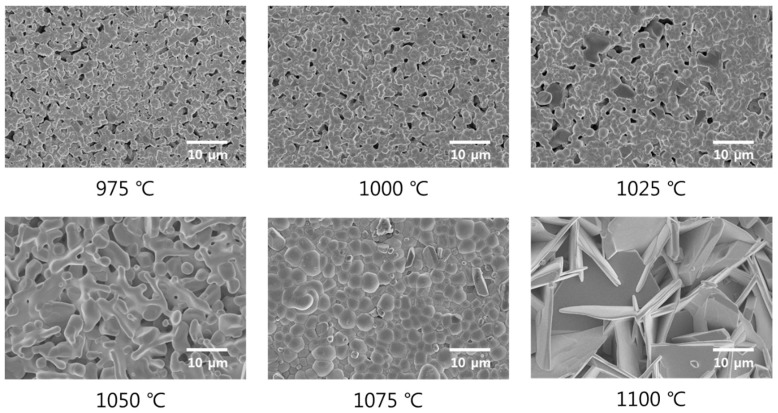
Field-emission scanning electron microscopy (FESEM) images of Bi(Mg_0.5_Ti_0.5_)O_3_–PbTiO_3_ according to sintering temperature.

**Figure 5 sensors-19-02115-f005:**
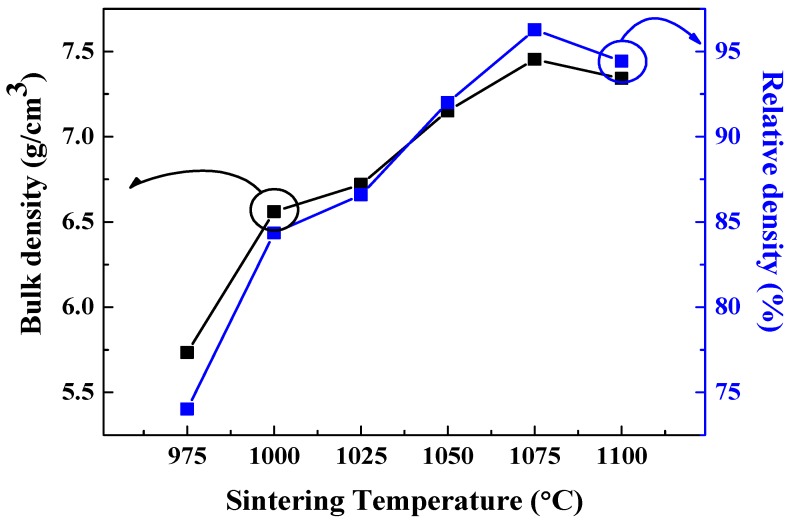
Bulk density and relative density of Bi(Mg_0.5_Ti_0.5_)O_3_–PbTiO_3_ ceramics according to sintering temperature.

**Figure 6 sensors-19-02115-f006:**
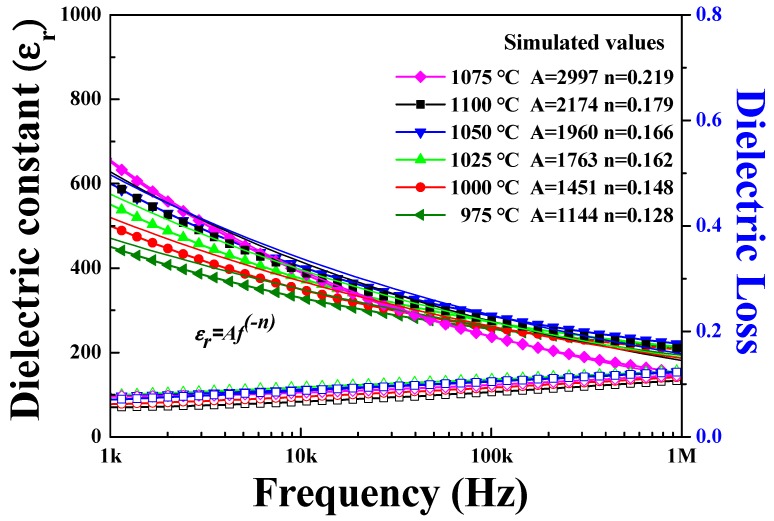
Frequency-dependent dielectric permittivity of Bi(Mg_0.5_Ti_0.5_)O_3_–PbTiO_3_ ceramics according to sintering temperature.

**Figure 7 sensors-19-02115-f007:**
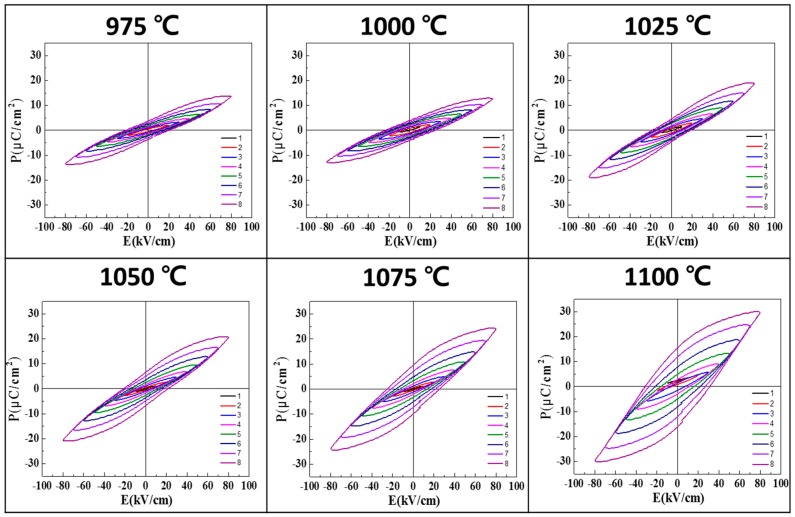
Electric field-dependent polarization properties of Bi(Mg_0.5_Ti_0.5_)O_3_–PbTiO_3_ according to sintering temperature.

**Figure 8 sensors-19-02115-f008:**
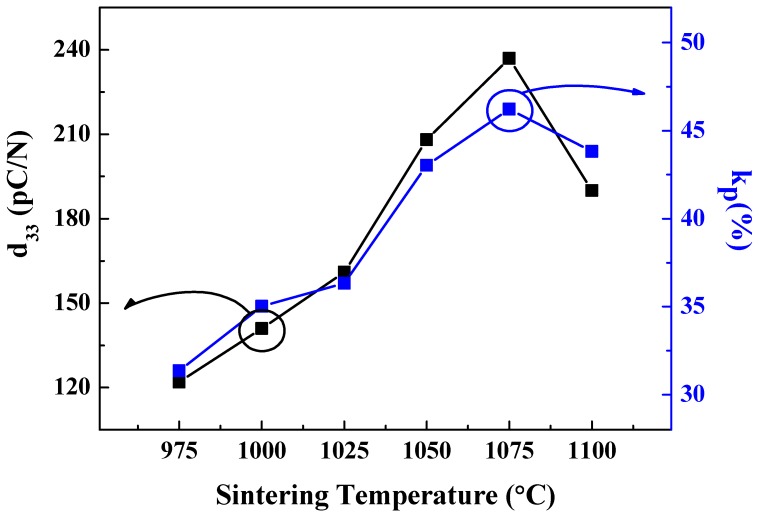
Piezoelectric coefficient d_33_ and k_p_ of Bi(Mg_0.5_Ti_0.5_)O_3_–PbTiO_3_ according to sintering temperature.

**Figure 9 sensors-19-02115-f009:**
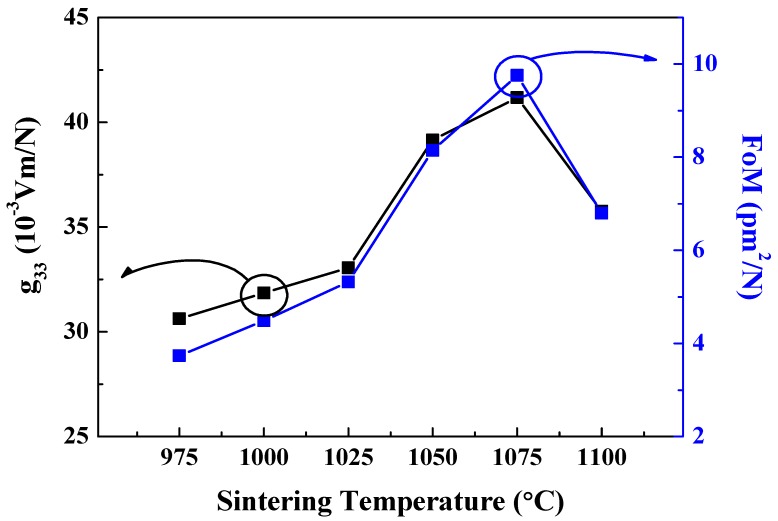
Piezoelectric voltage coefficient and figure of merit (FoM) of Bi(Mg_0.5_Ti_0.5_)O_3_–PbTiO_3_ according to sintering temperature.

**Figure 10 sensors-19-02115-f010:**
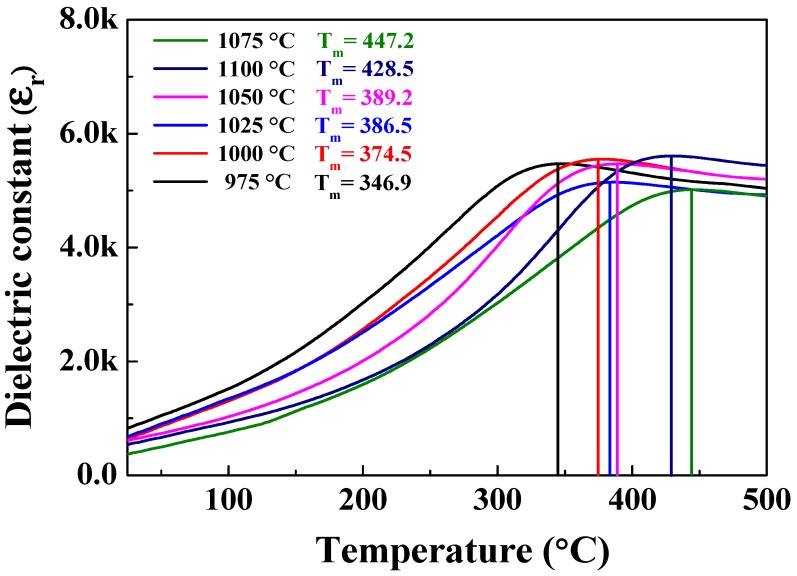
Temperature-dependent relative dielectric constant ε_r_ of Bi(Mg_0.5_Ti_0.5_)O_3_–PbTiO_3_ ceramic according to sintering temperature.

**Table 1 sensors-19-02115-t001:** Comparison of piezoelectric properties of BiMeO_3_–PbTiO_3_ and other lead content-reduced ceramics.

	Sintering Temperature	Dielectric Permittivity	Curie Temperature	Piezoelectric Charge Coefficient	Cost	Reference
BiFeO_3_–PbTiO_3_	1100 °C	500 for 0.3BF–0.7PT	500 °C	165 pC/N	Moderate	[12,16]
BiScO_3_–PbTiO_3_		1450 for 0.36BS–0.64PT	450 °C	460 pC/N	Very High	[12,17]
BiInO_3_–PbTiO_3_		600 for 0.1BiInO_3_–0.9TiO_3_	541 °C		Moderate	[12,18]
BiYbO_3_–PbTiO_3_	1140 °C	650 for0.1BY–0.9PT	590 °C	18 pC/N	Very High	[12,19]
BNT–PT0.55Bi(Ni_1/2_Ti_1/2_)O_3_–0.45PbTiO_3_	1050 °C		400 °C	293 pC/N	Moderate	[12,20]
BMT–PT(our samples)	1075 °C	625 for 0.62BMT–0.38PT	447 °C	237 pC/N	Moderate

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
