# Peer review of "Relaxation-Related Piezoelectric and Dielectric Behavior of Bi(Mg,Ti)O3–PbTiO3 Ceramic"

_sensors, 2019, doi:10.3390/s19092115_

Round 1
Reviewer 1 Report
In this manuscript, the authors investigated the effects of sintering temperature on the crystalline phase, microstructure, dielectric, ferroelectric and piezoelectric properties of 0.62Bi(Mg0.5Ti0.5)O3-0.38PbTiO3 ceramic. However, considering the quality of the journal, I will not suggest the publication of this manuscript in the Sensors mainly for the following issues:
(1) This paper is lack of novelty. There are several other similar articles [like J. Am. Ceram. Soc., 93 [10] 3330–3334 (2010)]. Since many systematic studies have been done, I cannot understand the motivation of this work as shown in your introduction part.
(2) The authors claimed that “Owing to this high relaxation behavior, 0.62BMT-0.38PT ceramics sintered at 1075 °C have the highest piezoelectric properties”, but the authors did not clarify the inner relationship between relaxation behavior and piezoelectric properties in this manuscript.
(3) As discussed in Introduction part, 0.62Bi(Mg0.5Ti0.5)O3-0.38PbTiO3 is in the MPB region. The authors should give the detailed crystalline structures when discussing the XRD results, such as rhombohedral or tetragonal structures.
(4) In the Experimental part, the authors should provide all the name of measurement apparatus.
(5) Check the text carefully and thoroughly revise and polish the languages.
Author Response
Reviewer1
Comments and Suggestions for Authors
In this manuscript, the authors investigated the effects of sintering temperature on the crystalline phase, microstructure, dielectric, ferroelectric and piezoelectric properties of 0.62Bi(Mg0.5Ti0.5)O3-0.38PbTiO3 ceramic. However, considering the quality of the journal, I will not suggest the publication of this manuscript in the Sensors mainly for the following issues:
Question (1) This paper is lack of novelty. There are several other similar articles [like J. Am. Ceram. Soc., 93 [10] 3330–3334 (2010)]. Since many systematic studies have been done, I cannot understand the motivation of this work as shown in your introduction part.
Answer 1), Yes, we have already checked this reference journal at the first stage. This journal mainly concerned with composition of BMT-PT and their related piezoelectric properties. The data and contents are totally different. The main differences can be listed as follows: a) we have performed XRD analysis data depending on the sintering temperature range, b) we performed with lattice parameter analysis and compared with tetragonality properties depending on the sintering temperature range. c) SEM analysis data depending on the sintering temperature range. d) We have polarization versus electric field data, all data which have mentioned are only belong to us. These two papers are totally different papers.
Question (2) The authors claimed that “Owing to this high relaxation behavior, 0.62BMT-0.38PT ceramics sintered at 1075 °C have the highest piezoelectric properties”, but the authors did not clarify the inner relationship between relaxation behavior and piezoelectric properties in this manuscript.
Answer 2), Yes, we believe that relation behavior of dipoles in 0.62BMT-0.38PT ceramics sintered at 1075°C have influence the dielectric and piezoelectric properties. Therefore, these following sentences are added in the manuscript.
‘The high exponent value of 0.219 for power law equation means that many different dipoles involved the relation process with continuous freezing out of dipoles with increasing frequency range. Therefore, 0.62BMT-0.38PT piezoelectric ceramics showed rapid change of dielectric constant by increasing the frequency range.’
Question (3) As discussed in Introduction part, 0.62Bi(Mg0.5Ti0.5)O3-0.38PbTiO3 is in the MPB region. The authors should give the detailed crystalline structures when discussing the XRD results, such as rhombohedral or tetragonal structures.
Answer 3) Yes, we believe that 0.62Bi(Mg0.5Ti0.5)O3-0.38PbTiO3 ceramics have mixture of rhombohedral and tetragonal structure phases. Therefore, the following sentences are added in the manuscript.
‘In our assumption, we believe 0.62Bi(Mg0.5Ti0.5)O3–0.38PbTiO3 ceramics have mixture of rhombohedra and tetragonal structure. This assumption will be analyzed and discussed after the XRD analysis’
Question (4) In the Experimental part, the authors should provide all the name of measurement apparatus.
Answer 4)
Yes, we have added detail expression for the measurement apparatus. The following expressions were added in the manuscript.
‘The crystalline properties were analyzed by X-ray diffraction method with CuKa radiation souorce ( Rruker -AXS) and the electric properties were analyzed by an HP 4294 impedance analyzer. Field emission scanning electron microscopy (FESEM) was used to examine the microstructure. The piezoelectric charge coefficient and electric field dependent polarization processes were performed in this experiment by employing Sayer Tower method with 0.1 Hz to know its piezoelectric and dielectric properties.’
Question (5) Check the text carefully and thoroughly revise and polish the languages.
Answer 5) Yes, we have consulted English Language Editing Company ‘Editages’ The following figure is the proof of editing process. English Language Editing Company

Reviewer 2 Report
Comments to the authors:
The manuscript is well written but it needs some corrections.
In the abstract should authors explain more what is new.
Page 1: 1. Introduction: Line 24: Authors should include: “The Pb(Zr,Ti)O3 is important also in high precision new switchable measurement methods where dielectric properties is highly important. These methods compensate environment effect, voltage offset, frequency drift, and temperature influence such as we can see in ref.: “
- Matko V. and Milanović M., Temperature-Compensated Capacitance-Frequency Converter with High Resolution, Sens. Actuators A, 220, 2014, 262-269, doi: 10.1016/j.sna.2014.09.022.
http://www.sciencedirect.com/science/article/pii/S0924424714004178
- Matko V., Next generation AT-cut quartz crystal sensing devices. Sensors, 2011, vol. 5, 11, 4474-4482, doi: 10.3390/s110504474.
- Nie, Jing; Liu, Jia; Li, Ning; Meng, Xiaofeng. Dew point measurement using dual quartz crystal resonator sensor Sensors & Actuators: B. Chemical. July 2017 246:792-799 Language: English. DOI: 10.1016/j.snb.2017.02.166
Authors should include the references above into the manuscript.
Check all pages (guideline for manuscript preparation): ferroelectricity. (1-3) For ………
Page 8: 4. Conclusion: What are main advantages of the 0.62BMT-0.38PT material, and where can be used?
Author Response
Manuscript code No : Sensors - 486839
Title: Relaxation related piezoelectric and dielectric behavior of Bi(Mg,Ti)O3-PbTiO3 ceramic
Author : Min Young Park1, Jae-Hoon Ji1, Jung-Hyuk Koh1,*
Dear Reviewer
We appreciate your detailed reviews of the manuscript entitled: ‘Relaxation related piezoelectric and dielectric behavior of Bi(Mg,Ti)O3-PbTiO3 ceramic’ . We modified our manuscript according to the reviewer’s comments and common guidance. Reviewer’s comment was very advisable and helpful for our manuscript to be better. For illuminating, overall manuscript was carefully revised. Manuscript was wholly revised like re-writing. The English expression of this manuscript was consulted by the English Language Editing Company ‘Editage’. All modified contents were highlighted. We have modified the manuscript accordingly, and detailed corrections are listed below point by point:
Comments to the authors:
The manuscript is well written but it needs some corrections.
Question (1) In the abstract should authors explain more what is new.
Answer 1) Yes, we have added new content to the abstract. Therefore, these following sentences are added in the manuscript.
‘Very high Curie temperature of 447 was recorded by 0.62BMT-0.38PT ceramics sintered at 1075 C. For the first time, we found that the origin of high Curie temperature, d33, and dielectric constant are relaxation behavior of different dipoles in 0.62BMT-0.38PT ceramics.’
Question (2) Page 1: 1. Introduction: Line 24: Authors should include: “The Pb(Zr,Ti)O3 is important also in high precision new switchable measurement methods where dielectric properties is highly important. These methods compensate environment effect, voltage offset, frequency drift, and temperature influence such as we can see in ref.: “
- Matko V. and Milanović M., Temperature-Compensated Capacitance-Frequency Converter with High Resolution, Sens. Actuators A, 220, 2014, 262-269, doi: 10.1016/j.sna.2014.09.022.
http://www.sciencedirect.com/science/article/pii/S0924424714004178
- Matko V., Next generation AT-cut quartz crystal sensing devices. Sensors, 2011, vol. 5, 11, 4474-4482, doi: 10.3390/s110504474.
- Nie, Jing; Liu, Jia; Li, Ning; Meng, Xiaofeng. Dew point measurement using dual quartz crystal resonator sensor Sensors & Actuators: B. Chemical. July 2017 246:792-799 Language: English. DOI: 10.1016/j.snb.2017.02.166
Authors should include the references above into the manuscript.
Answer 2) Yes, we have included the reviewer’s comments in the manuscript and added reference papers in the manuscript. Therefore, these following sentences are added in the manuscript.
‘The Pb(Zr,Ti)O3 is important also in high precision new switchable measurement methods where dielectric properties is highly important. These methods compensate environment effect, voltage offset, frequency drift, and temperature influence such as we can find the earlier reports. [2-4]’
‘Matko, V.; Milanovic, M. Temperature-compensated capacitance-frequency converter with high resolution, sensors and Actuators A 2014, 220, 262-269.
Matko, V. Next Generation AT-Cut Quartz Crystal Sensing Devices, Sensors 2011, 11, 4474-4482.
Nie, J.; Liu, J.; Li, N.; Meng, X. Dew point measurement using dual quartz crystal resonator sensor, Sensors and Auctuators B 2017, 246, 792-799.’
Question (3) Check all pages (guideline for manuscript preparation): ferroelectricity. (1-3) For ………
Answer 3) Yes, we have checked again the guideline for the manuscript preparation.
Question (4) Page 8: 4. Conclusion: What are main advantages of the 0.62BMT-0.38PT material, and where can be used?
Answer 4) Yes, we believe this material can be applied for the piezoelectric devices under harsh environment because it has reduced lead contents and high piezoelectric charge coefficient with lowered Curie temperature range of 447 °C. The following sentences were added in the manuscripts.
‘Very high Curie temperature of 447 was recorded by 0.62BMT-0.38PT ceramics sintered at 1075 °C. Since 0.62BMT–0.38PT piezoelectric ceramics have lowered lead content with high piezoelectric charge coefficient of 237 pC/N. Therefore, this material can be applied for the piezoelectric applications in the harsh environment’

Reviewer 3 Report
In this paper Authors decreased lead content of piezoelectric materials was studied and discussed. Authors have presented the results of studies that show that even though the lead content is decreased in Bi(Mg0.5Ti0.5)O3–PbTiO3 ceramics, it shows piezoelectric properties similar to that of lead zirconate titanite (PZT) based materials.
Below I presented some remarks that came to my mind during reading.
General remarks:
1. Line 28: It seems to me that the new sentence "Recently ..." is worth starting with a new paragraph.
2. Line 41: It seems to me that the new sentence "In this research…“ is worth starting with a new paragraph.
3. The introduction should be strongly quoted. In the introduction, the authors should present other works related to the topic of this paper. In my opinion, 6 references are not enough for a scientific paper. I suggest extending the introduction with other works.
4. Paper should be written in an impersonal form.
5. The presented paper lacks confrontation of the obtained results with other research. I suggest adding confrontation of the obtained results with other recent research. Such comparison significantly raises the meaning of the presented paper. I recommend Authors to conduct suitable comparisons to solve this issue. The Authors should also highlight what are the advantages and disadvantages when comparing their devised solution with other solutions from the scientific literature.
6. The Authors should present the findings also highlighting current limitations of their study, and briefly mention some precise directions that they intend to follow in their future research work.
7. In Conclusions, in addition to summarizing the action taken and results, the Authors should explain their significance. They should discuss in most of the obtained results.
8. Add the “Authors' contribution”.
9. References should be prepared in accordance with the Sensors template.
Author Response
Manuscript code No : Sensors - 486839
Title: Relaxation related piezoelectric and dielectric behavior of Bi(Mg,Ti)O3-PbTiO3 ceramic
Author : Min Young Park1, Jae-Hoon Ji1, Jung-Hyuk Koh1,*
Dear Reviewer
We appreciate your detailed reviews of the manuscript entitled: ‘Relaxation related piezoelectric and dielectric behavior of Bi(Mg,Ti)O3-PbTiO3 ceramic’ . We modified our manuscript according to the reviewer’s comments and common guidance. Reviewer’s comment was very advisable and helpful for our manuscript to be better. For illuminating, overall manuscript was carefully revised. Manuscript was wholly revised like re-writing. The English expression of this manuscript was consulted by the English Language Editing Company ‘Editage’. All modified contents were highlighted. We have modified the manuscript accordingly, and detailed corrections are listed below point by point:
Comments and Suggestions for Authors
In this paper Authors decreased lead content of piezoelectric materials was studied and discussed. Authors have presented the results of studies that show that even though the lead content is decreased in Bi(Mg0.5Ti0.5)O3–PbTiO3 ceramics, it shows piezoelectric properties similar to that of lead zirconate titanite (PZT) based materials.
Below I presented some remarks that came to my mind during reading.
General remarks:
Question (1) Line 28: It seems to me that the new sentence "Recently ..." is worth starting with a new paragraph.
Answer 1) Yes, we modified it to a new paragraph as you said.
Question (2) Line 41: It seems to me that the new sentence "In this research…“ is worth starting with a new paragraph.
Answer 2) Yes, we modified it to a new paragraph as you said.
Question (3) The introduction should be strongly quoted. In the introduction, the authors should present other works related to the topic of this paper. In my opinion, 6 references are not enough for a scientific paper. I suggest extending the introduction with other works.
Answer 3) Yes, we have added 8 more related references, as the reviewer said.
The added reference papers are as follows;
Wang, L.; Song, T.K.; Lee, S.C.; Cho, J.H.; Sakka, Y. Effect of Bi(B)O3 perovskite substitution on enhanced tetragonality and ferroelectric transition temperature in Pb(Zr,Ti)O3 ceramics, Mater Chem Phys 2011, 129, 322-325.
Matko, V.; Milanovic, M. Temperature-compensated capacitance-frequency converter with high resolution, sensors and Actuators A 2014, 220, 262-269.
Matko, V. Next Generation AT-Cut Quartz Crystal Sensing Devices, Sensors 2011, 11, 4474-4482.
Nie, J.; Liu, J.; Li, N.; Meng, X. Dew point measurement using dual quartz crystal resonator sensor, Sensors and Auctuators B 2017, 246, 792-799.
Li, J-F.; Wang, K. Ferroelectric and Piezoelectric Properties of Fine-Grained Na0.5K0.5NbO3 Lead-Free Piezoelectric Ceramics Prepared by Spark Plasma Sintering, J Am Cream Soc 2006, 89, 706-709.
Takenaka, T.; Nagata, H. Current status and prospects of lead-free piezoelectric ceramics, J Eur Ceram Soc 2005, 25, 2693-2700.
Baettig, P.; Schelle, C.F.; LeSar, R.; Waghmare, U.V.; Spaldin, N.A. Theoretical Prediction of New High-Performance Lead-Free Piezoelectrics, Chem Mater 2005, 17, 1376-1380.
Saleem, M.; Hwan, L.D.; Kim, I-S.; Kim, M-S.; Maqbool, A.; Nisar, U.; Pervez, S.A.; Farooq, U.; Farooq, M.U.; Khalil, H.M.W.; Jeong, S-H. Revealing of Core Shell Effect on Frequency-Dependent Properties of Bi-based Relaxor/Ferroelectric Ceramic Composites, Sci. Rep 2018, 8, 14146.
Question (4) Paper should be written in an impersonal form.
Answer 4) Yes, we have corrected the manuscript as the reviewer’s comments. Also the manuscript was consulted by English editing company ‘Editages’, which is very famous for the scientific writing.
Question (5) The presented paper lacks confrontation of the obtained results with other research. I suggest adding confrontation of the obtained results with other recent research. Such comparison significantly raises the meaning of the presented paper. I recommend Authors to conduct suitable comparisons to solve this issue. The Authors should also highlight what are the advantages and disadvantages when comparing their devised solution with other solutions from the scientific literature.
Answer 5) Yes, we have collects related papers and then analyzed to compare our results. Finally, we have summarized results in the table 1. The advantage of BMT-PT can be summerized that it has high piezoelectric properties with high Curie temperature and also very reasonable price. However, as count part, BiScO3-PbTiO3 have very high piezoelectric coefficient and high Curie temperature, but this material is very expansive to apply electronic device applications. As a result, this table was added in the manuscript and following sentence was added in the manuscript.
Table 1. comparison of piezoelectric properties of BiMeO3-PbTiO3 and other lead reduced ceramics.
Sintering temp | Dielectric permittivity | Curie temperature | Piezoelectric charge coefficient | Cost | reference | |
BiFeO3-PbTiO3 | 1100 °C | 500 for 0.3BF-0.7PT | 500 °C | 165 pC/N | moderate | 12,17 |
BiScO3-PbTiO3 | 1450 for 0.36BS-0.64PT | 450 °C | 460 pC/N | Very High | 12,18 | |
BiInO3-PbTiO3 | 600 for 0.1BiInO3-0.9TiO3 | 541 °C | moderate | 12,19 | ||
BiYbO3-PbTiO3 | 1140 °C | 650 for 0.1BY-0.9PT | 590 °C | 18 pc/N | Very High | 12,20 |
BNT-PT 0.55Bi(Ni1/2Ti1/2)O3–0.45PbTiO3 | 1050 °C | 400 °C | 293 pc/N | moderate | 12, 21 | |
BMT-PT (our samples) | 1075 °C | 625 for 0.62BMT-0.38PT | 447 °C | 237 pc/N | moderate |
‘The main advantage of BMT-PT piezoelectric system can be summarized in the table 1. As explained in the table 1, this system has high piezoelectric properties and high Curie temperature with moderate price. However, as a representative lead content reduced material, BiScO3-PbTiO3 have high piezoelectric coefficient and Curie temperature, but this material is very expansive to apply electronic device applications’
Question (6) The Authors should present the findings also highlighting current limitations of their study, and briefly mention some precise directions that they intend to follow in their future research work.
Answer 6) Yes we have addressed current limitations for the these (1-x)BiMeO3–xPbTiO3 (Me3+ = Fe, Sc, Mg, In, Y, Yb, Ga), based materials and presented research direction of these materials for future applications. Therefore, the following texts were addressed in the manuscript.
‘The multiferroic and relatively weak ferroelectric properties in BiMeO3–PTiO3 system, which usually observed in the electric field dependent polarization process can be main obstacles for the future piezoelectric applications. However, as we have mentioned before, relatively high piezoelectric charge coefficient more than 200 pC/N with higher Curie temperature range more than 400 °C can bright the future applications for the multifunctional applications including actuators and sensors’
Question (7) In Conclusions, in addition to summarizing the action taken and results, the Authors should explain their significance. They should discuss in most of the obtained results.
Answer 7) Yes, we have added outstanding results in the conclusion. Therefore, these following sentences are added in the manuscript.
‘Very high Curie temperature of 447 was recorded by 0.62BMT-0.38PT ceramics sintered at 1075 °C. Since 0.62BMT–0.38PT piezoelectric ceramics have lowered lead content with high piezoelectric charge coefficient of 237 pC/N. Therefore, this material can be applied for the piezoelectric applications in the harsh environment.’
‘By reducing the lead content of piezoelectric materials, it is expected not only to help prevent environmental pollution on the earth, but also to be more useful as a piezoelectric material because of its excellent characteristics compared with lead-free piezoelectric material. It will be used for piezoelectric transducers or sintering aid using ferroelectricity that does not change to high temperature.'
Question (8) Add the “Authors' contribution”.
Answer 8) Yes, we have corrected the authors contribution. Therefore, these following sentences are added in the manuscript.
‘Author Contributions: M.-Y. Park performed the experiments and wrote the paper, J-H. Ji and J.-H. Koh analyzed experimental data.’
Question (9) References should be prepared in accordance with the Sensors template.
Answer 9) Yes, we have modified the above references to fit the sensor template, as the reviewer said.

Round 2
Reviewer 1 Report
The analysis and description are adequate for a publication, and the results and conclusions are convincing. The revised manuscript has been improved according to the referee’s comments and can be accepted in the present format.
Reviewer 3 Report
The comments and corrections made as part of my first review have been addressed. The authors have taken into consideration all my questions and comments. I recommend to accept the new version of the paper.